# Recent Advances in Metal-Based NanoEnhancers for Particle Therapy

**DOI:** 10.3390/nano13061011

**Published:** 2023-03-10

**Authors:** Yao-Chen Chuang, Ping-Hsiu Wu, Yao-An Shen, Chia-Chun Kuo, Wei-Jun Wang, Yu-Chen Chen, Hsin-Lun Lee, Jeng-Fong Chiou

**Affiliations:** 1Department of Radiation Oncology, Taipei Medical University Hospital, Taipei 110301, Taiwan; 215123@h.tmu.edu.tw (Y.-C.C.);; 2Department of Radiology, School of Medicine, College of Medicine, Taipei Medical University, Taipei 110301, Taiwan; 3Proton Center, Taipei Medical University Hospital, Taipei Medical University, Taipei 110301, Taiwan; 4Department of Pathology, School of Medicine, College of Medicine, Taipei Medical University, Taipei 110301, Taiwan; 5Graduate Institute of Clinical Medicine, College of Medicine, Taipei Medical University, Taipei 110301, Taiwan; 6International Master/Ph.D. Program in Medicine, College of Medicine, Taipei Medical University, Taipei 110301, Taiwan; 7School of Health Care Administration, College of Management, Taipei Medical University, Taipei 110301, Taiwan; 8Ph.D. Program for Cancer Molecular Biology and Drug Discovery, College of Medical Science and Technology, Taipei Medical University, Taipei 11031, Taiwan

**Keywords:** particle therapy, proton therapy, radiosensitization, radioresistance, nanomedicine, theranostic, nanoparticles

## Abstract

Radiotherapy is one of the most common therapeutic regimens for cancer treatment. Over the past decade, proton therapy (PT) has emerged as an advanced type of radiotherapy (RT) that uses proton beams instead of conventional photon RT. Both PT and carbon-ion beam therapy (CIBT) exhibit excellent therapeutic results because of the physical characteristics of the resulting Bragg peaks, which has been exploited for cancer treatment in medical centers worldwide. Although particle therapies show significant advantages to photon RT by minimizing the radiation damage to normal tissue after the tumors, they still cause damage to normal tissue before the tumor. Since the physical mechanisms are different from particle therapy and photon RT, efforts have been made to ameliorate these effects by combining nanomaterials and particle therapies to improve tumor targeting by concentrating the radiation effects. Metallic nanoparticles (MNPs) exhibit many unique properties, such as strong X-ray absorption cross-sections and catalytic activity, and they are considered nano-radioenhancers (NREs) for RT. In this review, we systematically summarize the putative mechanisms involved in NRE-induced radioenhancement in particle therapy and the experimental results in in vitro and in vivo models. We also discuss the potential of translating preclinical metal-based NP-enhanced particle therapy studies into clinical practice using examples of several metal-based NREs, such as SPION, Abraxane, AGuIX, and NBTXR3. Furthermore, the future challenges and development of NREs for PT are presented for clinical translation. Finally, we propose a roadmap to pursue future studies to strengthen the interplay of particle therapy and nanomedicine.

## 1. Introduction

Cancer is a common disease and remains a leading cause of death globally in the 21st century. According to estimates from the World Health Organization in 2022, cancer will be the first or second leading cause of death worldwide, with more than 18 million diagnosed cases accounted for and nearly 10 million deaths in 2020 [1]. The global cancer burden is expected to increase to 28.4 million new cancer diagnoses by 2040, which represents a 47% rise in number of cases from 2020 [2]. Radiotherapy (RT) has become one of the main treatment modalities for cancer over the past century with more than two-thirds of the patients receiving RT alone or along with combinations of other treatments [3]. During RT, the photon beam is mainly used to damage cancer cells without the need to perform surgical incisions on the patient body. This non-invasive method is used to effectively treat deeply seated tumors; however, as the interaction of photon beams and human tissue is irreversible, photon RT not only deposits most of its energy to the target cancer cells tumors, but also damages healthy tissues along its path (Figure 1). With the rapid development and improvement in technology, various radiotherapeutic strategies are developed to improve the precision of RT, such as conformal radiotherapy (CFRT), intensity-modulated radiation therapy (IMRT), and particle therapy (e.g., proton therapy (PT), carbon-ion beam therapy (CIBT), and boron neutron capture therapy (BNCT)). Compared with photon beam RT, PT produces a characteristic depth dose curve. At the end of the particle’s range, the radiation dose is released rapidly and reaches a peak, known as the Bragg peak (Figure 1A). In addition, the initial beam can be tuned using energy selection system modulation to alter the beam energy to create a uniform shape and dose distribution to encompass the treatment of lesions, thus producing a spread-out Bragg peak (SOBP) (Figure 1B) [4,5]. SOBP imparts a good dosimetric distribution to particle therapy, which maximizes the efficiency of killing of tumor tissues and protects surrounding normal tissues around to a great extent [6]. Particle therapy is thus considered, at least for a number of indications, superior to conventional RT [7,8,9]. Hung et al. compared clinical outcomes of proton versus photon ablative radiation therapy in patients with unresectable hepatocellular carcinoma (HCC). In their study, treatment with proton radiation therapy showed improved overall survival and decreased incidence of non-classic radiation-induced liver disease compared with photon radiation therapy. Consistent with their results, Huang’s group compared the clinical outcomes of HCC patients, those treated with either PT or RT, and determined that the photon group had better overall survival rate. Even though clinical evidence demonstrating the benefit of protons over photons is still limited, according to the data published by the Proton Therapy Co-Operative Group (https://www.ptcog.ch/index.php/facilities-in-operation-restricted, accessed on 31 December 2022), particle therapy treatment is available at 121 facilities worldwide, including 14 CIBT centers in operation. In 2022, six PT facilities (e.g., one in Anhui (China), one in Kanagawa (Japan), one in Taipei (Taiwan), one in Bangkok (Thailand), one in Missouri (USA) and one in Pennsylvania (USA)) as well as one CIBT facility in Taipei (Taiwan) have commenced operation. As of 2022, over 321,000 patients have been treated with particle therapy (PT and CIBT).

Although particle therapy has significant advantages over conventional RT for treating cancer by reducing the radiation damage to surrounding normal tissues, the radiation dose deposited by the particle beam in normal tissues which locates in the path before the tumor is unavoidable, which may cause normal tissue damage [10]. Many strategies have been suggested to balance treatment outcome of conventional RT with side effects, such as increasing radiation resistance in healthy tissues by using a very high-dose-rate (>40 Gy/s) RT (known as FLASH RT) [11,12], reversing tumor tissue radiation resistance using synergistic therapeutics [13,14,15,16], increasing radiation sensitization, and limiting radiation dose deposition in the tumor using nanosensitizing agents [17,18,19]. Of these, metal-based nanomaterials or “nano-radioenhancers” (NREs) have attracted significant attention in radiation oncology because of their strong photoelectric absorption coefficients for high atomic number (Z) metallic elements [20]. In addition, the tumor microenvironment differs from normal tissue and can be exploited for enhanced drug delivery. For example, in tumoral regions, leaky vasculature with large pores (100 nm to 2 μm in diameter) and inefficient lymphatic drainage facilitate passive NPs accumulation through Enhanced Permeability and Retention (EPR) effect. In comparison with the nonspecific EPR effect, active targeting NPs take advantage of the site-specific ligands to specifically bind to surface receptors expressed on target cell membranes that lead to the internalization of nanoparticles via receptor-mediated endocytosis, thereby enhancing the therapeutic effects [21,22,23]. Until recently, the preclinical results of nanoparticles (NPs) combined with RT have resulted in multiple clinical trials (Table 1). While nanosensitizing agents are now undergoing a clinical stage with conventional RT, only a handful of studies have been dedicated to evaluating the combination of metal-based NREs with particle therapy. Although a number of reviews regarding this topic have been published, the reviews have focused almost exclusively on photon irradiation (e.g., low-energy (keV) and high-energy (MeV) photons) [24,25,26]. In fact, since the physical mechanism is different from photon RT and particle therapy [27,28], there has been an increasing number of studies focused on the potential of particle therapy combined with specific NREs, particularly involving gadolinium, and hafnium. Therefore, this topical review is motivated by recent developments using NREs to enhance particle therapy in preclinical and clinical applications. The review is structured as follows: Section 2 sheds light on possible mechanisms and predicts the impact of ion beams and NP characteristics; Section 3 reviews progresses made to date from multiscale preclinical to clinical studies that have contributed to an increased interest in NREs for enhancing particle therapy; and Section 4 summarizes the challenges and promising opportunities in the translation of nanotechnology to incident particle therapies from our review along with our own perspective on the outlook for NP-enhanced radiation therapies.

## 2. Mechanisms of Radiosensitization by Nanomaterials

The basic rationale for using nanomaterials as NREs results from the physical dose enhancement that occurs following radiochemical and biological reactions in the targeted tissue [39]. The physical dose enhancement is caused by the generation of secondary X-rays, photoelectrons, and Auger electrons (Figure 2). The biochemical steps include oxidative stress and reactive oxygen species (ROS) production, DNA damages, reduced repair, cell cycle arrest, and bystander effects within the tissue [40]. For photon irradiation, Hainfeld et al. first reported the use of 1.9 nm-diameter gold (Au, Z = 79) particles (AuNPs) as a therapeutic advantage when combined with 250 kVp X-rays. This in vivo study demonstrated tumor regression with a significantly increased one-year survival rate of 20% when the tumor was treated with 250 kVp X-rays alone to 86% for the combination. Subsequently, publications involving AuNPs in cancer RT grew rapidly because of the high X-ray absorption coefficient and ease of synthetic manipulation, which enabled the particle’s physicochemical properties to be controlled with better precision [41,42]. In the coming years, other sophisticated NPs composed of heavy elements, such as titanium (Ti, Z = 22) [42,43], iodine (I, Z = 53) [44,45], gadolinium (Gd, Z = 64) [46,47,48], hafnium (Hf, Z = 72) [49,50,51], and bismuth (Bi, Z = 83) [52,53], have been studied extensively. Based on these studies, kV X-rays as a radiation modality can use the high photoelectric absorption cross-section and the consecutive release of secondary electrons (including Auger electrons and photoelectrons) from NREs as an advantage to further enhance the radiation effects(Figure 3A). In contrast to the photoelectric effect, at clinically relevant high radiation energy (megavoltage, MV), Compton scattering becomes the dominant interaction process with NREs (Figure 3B). In view of implementing the combination of high Z NREs with megavoltage photons, Monte Carlo simulation has been performed to theoretically calculate the dose enhancement effect of different concentrations, sizes, and clustering of various NREs [54,55,56]. In addition, experimentally evaluated in vitro cell models and in vivo in tumor bearing animals have provided promising results, even when the enhancement factor for radiation sensitization is lower for MV photons compared with that of photons in the kV range [31,57,58,59,60].

Although NP radioenhancement mechanisms with clinical MV photon beams are not well known, the use of charge particles in NP dose enhancement was not well studied. In 2010, Liu et al. measured the first radiosensitization effect resulting from metallic nanoparticles (MNPs) for cells irradiated with a proton beam. In this experiment, CT26 and EMT-6 cancer cells were irradiated from several radiation sources with different doses, including biological irradiator [E(average) = 73 keV], a Cu-Kalpha (1) X-ray source (8.048 keV), a monochromatized synchrotron source (6.5 keV), a radio-oncology linear accelerator (6 MeV), and a proton source (3 MeV). However, AuNP-induced radiosensitization was observed for most of the X-ray sources with varying energy spectrums, except the proton source. Although no statistical significance was reported, the results were rather disappointing; however, this experiment prompted further studies and measurements of AuNP radiosensitization in cells irradiated with a proton beam [62]. In the same year, Kim et al. was the first to show that MNPs prolong the life of mice treated with 45 MeV PT. CT-26-bearing mice were treated with a combination of gold or iron NPs, and the proton irradiation had significantly reduced tumor growth in the mice after irradiation compared with mice treated with only protons beams [63]. Subsequently, Li et al. performed an in vitro experiment to measure in survival fraction changes and examined the linear energy transfer (LET)-dependence of AuNP radiosensitization in PT. A marked radiosensitization effect of AuNPs was observed with 25 keV μm^−1^ protons, but not with 10 keV μm^−1^ protons [64]. This result is consistent with that of Cunningham et al., in which no significant increase in protons at low LET range was observed [17,65]. In addition, to confirm the radiosensitization mechanism when NPs are used in combination with irradiation, the free radical scavenger DMSO was used to reduce indirect damage resulting from ROS. It could be attributed to the fact that Coulomb collision with NPs ionized the atomic electrons to generate characteristic X-rays and Auger electrons and subsequently yield secondary water radiolysis, and also enhanced production of ROS [11,66,67] (Figure 3D and Figure 4A). In addition to PT, AuNPs exhibited a significant increase in radiosensitization with CIBT. Kaur et al. prepared AuNPs coated with glucose. The excellent targeting and internalization ability contributed to a significant increase in cellular uptake. Moreover, in vitro studies have shown that the combined treatment of Glu-AuNPs with a carbon-ion beam in HeLa cervical cancer cells may significantly improve anti-cancer efficacy and confirmed that AuNPs had a significant RT sensitization effect with CIBT [68]. Subsequently, Kim et al. reported that AuNPs were effective radiosensitizers when combined with neutron therapy against hepatocellular carcinoma. The sensitization enhancement ratio (SER) was 1.35–1.80 for hepatocellular carcinoma cells. Compared with γ-ray radiation, the exposure to neutron radiation with the use of AuNPs induced cell cycle arrest, DNA damage, and cell death to a greater extent, along with a marked suppression of cell migration and invasion. The results of the study by Kim et al. expanded the application range of NREs to neutron therapy [69]. In a recent study, Abdul Rashid et al. compared the molecular effects induced by Fe, Au, Pt, and Bi NPs under the exposure of 150 MeV protons. The induction of NER was amplified by all types of NPs. For protons at the center of the spread-out Bragg peak, the SER obtained by the inclusion of the superparamagnetic iron oxide NPs, SPIONs, AuNPs, PtNDs, and BiNRs was 1.95, 2.64, 3.08 and 4.93 respectively. Measurements of intercellular ROS for HCT 116 cells indicated that BiNRs produce ROS at 475% and PtNDs at 340%, followed by AuNPs and SPIONs at 230% compared with the control, which was consistent with the SER results (Figure 4B) [70]. Recently, multicomponent metallic NPs were used to improve RT for simulated proton irradiation against cancer cells. Klebowski et al. synthesized AuNPs decorated with PtNPs and PdNPs by a green chemistry method using gallic acid. The large surface of fancy shaped bimetallic NPs ensures a large contact area with the cells and results in an increased amount of ROS destruction in cancer cells [19]. The administration of 30 nm PtAuNPs and PdAuNPs as potential radiosensitizers in PT of colorectal cancer showed that this combined approach resulted in a significant inhibition of cancer cell proliferation and viability, whereas normal cells were less affected during treatment (Figure 4C). Subsequently, the same group designed two bimetallic PtPdNP structures, nano-alloy and core-shell, and checked the radiosensitizing properties. In this experiment, nano-alloy PtPdNPs exhibited superior radiosensitization in simulated proton irradiation compared with their PdPt core-shell counterparts, suggesting that the presence of Pd atoms on the surface of these NPs imparts better radiosensitizing effects than Pt atoms (Figure 4D) [71]. In addition to high Z elements, some NPs composed of elements with a relatively low atomic number are also efficient radiosensitizers [72,73,74,75]. For example, Guerreiro et al. performed a survey of 22 different metal oxide NPs and examined their intrinsic ability to generate ROS using clinically relevant megavoltage X-ray photons (6-MV X-rays). They determined that two metal oxides (V_2_O_5_, for hydroxyl radicals; TiO_2_ for superoxides) increased radiation-induced radical formation and hypothesized that the catalytic activity of the NP surface influenced the radioenhancer capability [76]. Recently, surface-catalyzed reactions of metal-based NPs have been recognized as important induction mechanisms of the ROS cascade during PT (Figure 5A). Zwiehoff et al. compared ROS generation between AuNPs and PtNPs as well as Au_90_Pt_10_ alloy NPs at an equivalent surface area to determine the material-specific differences and effects. The ROS generation of PtNPs significantly outperformed that of AuNPs and Au_90_Pt_10_ alloy NPs at the same alloy particle size and surface area atom concentration. These results indicate that the chemical reactivity of the NP surface is the main contributor to ROS generation during PT (Figure 5C,D) [77]. These observations are consistent with the results of Gerken et al., who showed that TiO_2_ NPs exhibit strong radiocatalytic activity in photons (150 kVp and 6 MV) and 100 MeV proton irradiation settings and resulted in the formation of hydroxyl radicals and nuclear interactions with protons (Figure 5B) [78]. Based on the results of these comprehensive studies, better predictions can be made regarding the optimization of NREs for PT and other ion beam therapies.

## 3. Clinical Trials Involving the Translation of Nanotechnology to Charge Particle Therapy

In recent years, the improvement and development of NPs has rapidly evolved, driven by the aim to overcome the limitations of free therapeutics, navigating biological barriers, and even the clinical failures of current drugs [80,81]. Over the last 20 years, approximately 80 nanomedicine products have been approved by the Food and Drug Administration (FDA) and the European Medicines Agency (EMA) to be available for marketing. Hensify^®^, NBTXR3, is currently the only FDA/EMA-approved nanosized radioenhancer for cancer RT [82,83]. Clinical trials are underway to evaluate the efficiency and safety of four NP candidates for both kV and MV photons utilizing gadolinium (Gd) chelates into polysiloxane NPs (AGuIX), hafnium-based NPs (NBTXR3, also known as PEP503), superparamagnetic iron oxide NPs (Ferumoxytol), and albumin-bound paclitaxel (Abraxane).

Ferumoxytol is a 17–30 nm iron oxide NP with a polyglucose sorbitol carboxymethylether coating. Recently, Ferumoxytol plus stereotactic body radiotherapy (SBRT) is being evaluated (NCT04682847, Phase I) for treatment of primary and metastatic hepatic cancers. The goal of the study is to develop an MRI-linac SPION-based RT planning platform for the detection and conformal avoidance of residual, functionally active hepatic parenchyma during liver SBRT for primary and metastatic hepatic malignancies in patients with hepatic cirrhosis.

Abraxane is an albumin-bound form of paclitaxel with a mean particle size of approximately 130 nanometers. The first Phase I trial (NCT00736619) included patients with Stage III-IVB head and neck squamous cell carcinoma (HNSCC) and therapy consisted of Abraxane + IMRT combined with cetuximab [36]. The goal of this study was to establish a safe dose range of albumin-bound paclitaxel in combination with cetuximab and RT. The success of the initial trials using Abraxane led to subsequent clinical trials, such as NCT03107182, a Phase II trial for patients with HNSCC. All enrolled patients will receive induction chemotherapy and will be assessed based on response to chemotherapy as well as high- or low-risk status. Subsequently, patients at low risk and those who do not qualify for TORS (due to volume of disease or poor visualization/access) or refuse TORS will be administered de-intensified treatment with radiation alone at 50 Gy. Next, an Abraxane Phase I trial (NCT02394535) was initiated to treat advanced pancreatic cancer patients with combined RT assisted by Abraxane and apecitabine orally twice daily on days 1–5 [38]. The initial efficacy results of concurrent Abraxane with capecitabine RT are promising; however, the Phase II trial (NCT01921751) was terminated because of an unreasonable timeframe [37].

Developed by Nanobiotix, NBTXR3 consists of crystalline hafnium oxide NPs functionalized by a negatively charged surface coating with a size centered at 50 nm [84]. Recently, several clinical trials were using NBTXR3 crystalline NPs in combination with RT. The initial Phase I study (NCT01433068) focused on locally advanced sarcoma and consisted of a single IT injection of NBTXR3 followed by RT (5 weeks, 50 Gy, 2 Gy/fraction) beginning 24 h following injection, with tumor resection at 6–8 weeks post irradiation. Additional Phase I/II trials and one Phase II/III trial followed, which included participants with various cancer types, such as head and neck cancer (squamous cell carcinoma) (NCT01946867, NCT04862455, NCT04892173 and NCT02901483), rectal cancer (NCT02465593) [35], hepatocellular carcinoma (liver cancer) (NCT02721056 and NCT05039632), prostate cancer (NCT02805894), adult soft tissue sarcoma (NCT02379845) [33], pancreatic cancer (NCT04484909), lung non-small cell carcinoma (NCT04505267), esophageal cancer (NCT04615013), and other advanced cancers (NCT03589339). The success of the clinical trials consisting of MV photon beams combined with NBTXR3 NREs led to further clinical trials. The design for the Phase II trial NCT04834349 included intratumoral/intranodal injection of NBTXR3 activated by intensity-modulated PT along with an anti-PD1 immunotherapy agent, pembrolizumab. Unfortunately, this trial was withdrawn because no participants were enrolled.

AGuIX are sub-5 nm NPs consisting of a polysiloxane matrix and gadolinium chelates. The first Phase I trial (NCT02820454) focused on multiple brain metastases and involved escalation of doses with an intravenous injection of 15, 30, 50, 75, and 100 mg/kg doses [29]. This was followed by whole brain RT (10 × 3 Gy over 3 weeks). AGuIX enables both radiosensitization and multimodal imaging of tumors prior to irradiation through the use of MRI, and CT, respectively. The metastases previously detected with a conventional T1 MRI contrast agent were also detected by AGuIX. This indicated that AGuIX can pass through the blood–brain barrier selectively in brain metastases, leading to the possibility of a differential response between the tumor and dose-limiting normal tissue. More Phase I/II trials followed, which included patients with various cancers, such as brain cancer (NCT03818386, NCT03308604, NCT04094077, and NCT04881032) [30,31,32], cervical cancer (NCT03308604), and other advanced cancers (NCT04899908, and NCT04789486). There are two ongoing clinical trials using polysiloxane Gd-chelate-based NPs (AGuIX) for hypofractionated PT (2020-003671-17/EMA, and NCT04784221/FDA). A Phase II trial is enrolling patients with recurrent cancer or a new tumor in an irradiated territory. The primary endpoint is to evaluate the efficacy of AGuIX activated by hypofractionated PT. Specifically, PT will be conducted for four consecutive weeks with five sessions per week (20 sessions) from D1 to D26. AGuIX will be injected intravenously on days 1, 8, and 15 during PT according to a standard regimen. This trial is a study that evaluates tumor regression and the local progression-free survival rate. Recently, a subsequent generation of AGuIX, Bi@AGuIX were obtained by grafting additional DOTA chelates onto AGuIX followed by the complexation of bismuth under acidic conditions. The addition of bismuth to AGuIX may lead to a greater radiosensitization effect and MRI and CT contrast enhancement [85]. Currently, Bi@AGuIX is undergoing preclinical evaluation.

## 4. Conclusions and Future Challenges

The development of NREs to improve particle therapy performance is only at early stage. Nevertheless, NREs have emerged over decades as a promising tool to address ionizing radiation. For clinical cancer treatment, a few products have been commercialized, such as AGuIX, NBTXR3, and Abraxane. The results obtained with several different NPs are very promising, but more effort is required to overcome the current challenges. Understanding key physical and biological parameters and underlying mechanisms of action involved in the radiosensitization effects of NPs remains an issue for developing new NREs. First, NPs with larger surface area provides better opportunities for surface interactions, which is especially important in the context of generating ROS [86,87]. Second, other considerations include their specific surface properties and diameters. Third, the interaction between NPs and cellular components should also be considered. The use of passive targeted MNPs results in inefficient accumulation in the tumor through EPR effect, which is insufficient to enhance radiosensitizing effect for cancer radiotherapy. According to the literature, the intratumour concentration of Au should be above 0.1 mg Au/mL for a radiosensitization effect in vitro [88,89]. Engineered MNPs that are functionalized with moieties able to actively target a tumor or its microenvironment have the potential to improve upon the efficacy and accuracy. Furthermore, cellular localization is necessary to achieve effective radiosensitization. ROS generated from the radiolysis of water deposits in the immediate vicinity of the NREs, considering their short life time in cells, the travelling distance of ROS is not more than 6 nm on average. Therefore, ROS formation close to cell nuclei or mitochondria may play a significant role in the radiosensitization capacity. In addition to physical and chemical mechanisms, biological effect contributes significantly to radiosensitization. Intracellular accumulation of NPs may affect antioxidant enzymes (such as thioredoxin reductase and glutathione peroxidase) expression and change the balance of oxidative stress. In addition, NPs sensitize cancer cells to radiation by causing cell cycle disruption, and DNA repair inhibition should not be ignored. Finally, current evaluation methods for biocompatibility are mostly focused on acute toxicity; however, the long circulation time of NPs and long-term reticuloendothelial system accumulation may cause problems associated with systemic toxicity and chronic immunogenic response. Therefore, detailed biocompatibility and toxicity studies of these MNPs are essential prior to their use in clinical practices.

The intention of this review was to provide an overview of the radiosensitization potential of MNPs in PT, which may help to address part of the current gap in the translation of these nanotechnologies into the clinic. We summarized the recent clinical trials involving MNP RT and elaborated on the radiosensitization mechanism. We anticipate that the continued efforts of all researchers in this work will result in the use of this knowledge to develop the next generation of NREs of cancer RT to improve cancer treatment.

## Figures and Tables

**Figure 1 nanomaterials-13-01011-f001:**
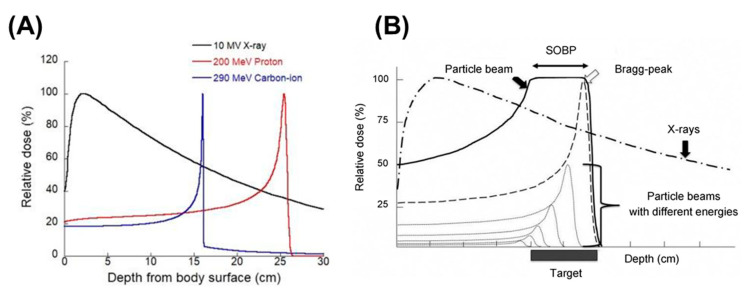
(**A**) Depth-dose distributions of clinical X-ray, proton, and carbon-ion beams, and (**B**) the spread-out Bragg peak (SOBP). Reproduced from [5], with permission from Elsevier, Copyright 2021.

**Figure 2 nanomaterials-13-01011-f002:**
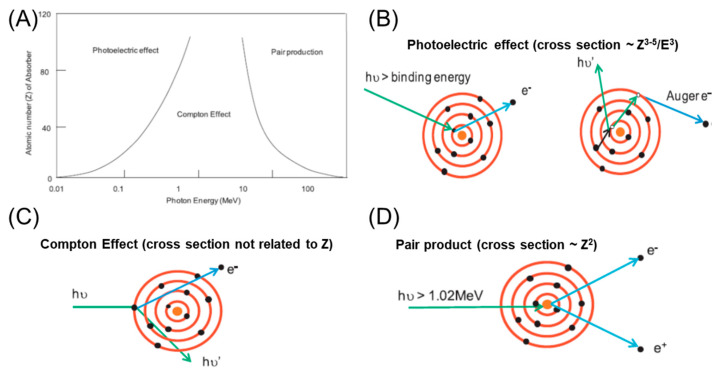
(**A**) Radiation energy and atomic number (Z) dependent interaction between radiation and Illustration of the (**B**) Photoelectric effect, (**C**) Compton Effect, and (**D**) pair production. (**B**) In the photoelectric effect (10–500 keV): the energy of the incident photon (hυ) is fully absorbed by an electron in the inner shell of an atom, and the electron is ejected from the atom. The vacant orbit is filled with an electron from an outer shell with high energy; extra energy is either released as a photon or absorbed by another electron in an outer shell, which is ejected from the atom (Auger electron). This Auger effect occurs in cascade if there are multiple shells of electrons in the atom. (**C**) In the Compton Effect (500 keV–1.02 MeV): the energy of the incident photon is partially absorbed by an electron in the outer shell of an atom, and the extra energy is released as photons. (**D**) In Pair product: when the energy of an incident photon is at least two-fold larger than mec2 (>1.02 MeV), and the energy is fully absorbed by the nucleus of an atom, a pair of electrons and positrons are generated from the nucleus. Adapted with permission [39]. Elsevier, Copyright 2017.

**Figure 3 nanomaterials-13-01011-f003:**
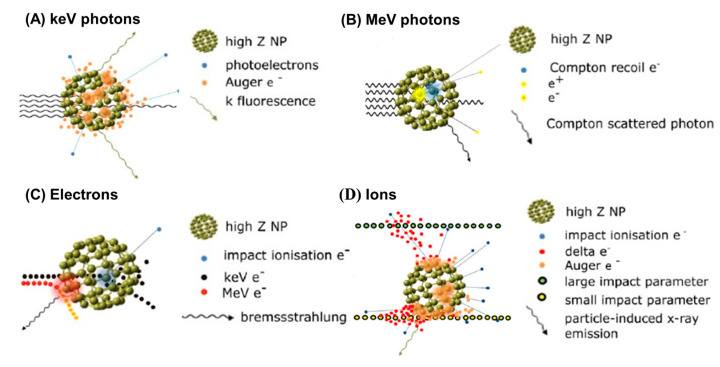
Schematic illustration of inelastic interactions with a high−Z NP for: (**A**) incident keV photons (orange clouds represent photoelectric events); (**B**) incident MeV photons (blue and yellow clouds represent Compton scatter and pair production events, respectively); (**C**) incident electrons (blue and red clouds represent large and small impact parameters leading to ionization and bremsstrahlung, respectively); and (**D**) incident ions (orange clouds indicate impact ionization events). Adapted with permission [61]. IOPSCIENCE, Copyright 2018.

**Figure 4 nanomaterials-13-01011-f004:**
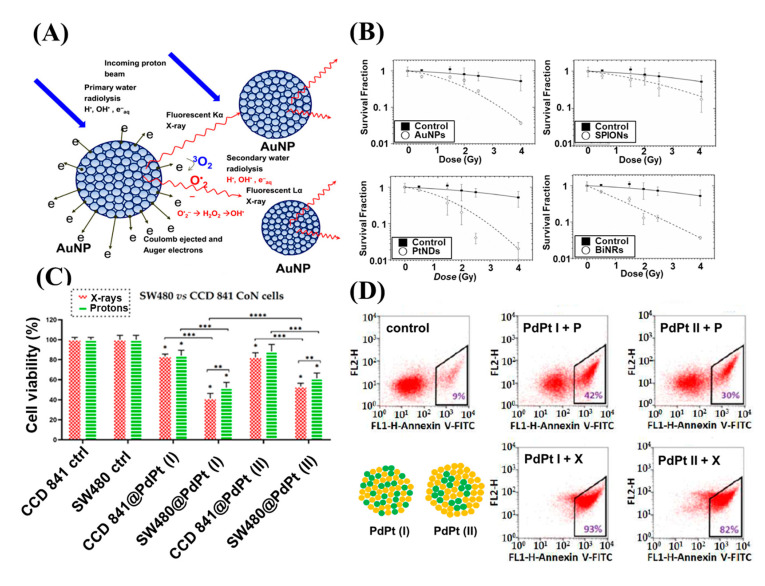
(**A**) A schematic diagram of low-energy electrons (LEEs) and enhanced ROS generation from proton-stimulated high-Z NPs via Coulomb decay path. (**B**) Survival curve as a function of dose for untreated cells w and cells loaded with NREs (AuNPs, SPIONs, PtNDs and BiNRs) irradiated with 150 MeV proton beam. (**C**) Viability of SW480 irradiated with X-ray (red bars) or proton (green bars), as well as CCD 841 CoN normal colon epithelium cells irradiated with X-ray (red bars) or proton (green bars) after their incubation with PdPt (I) or PdPt (II). (**D**) Apoptosis of SW480 determined by Annexin V-binding assay after the addition of PdPt NPs, followed by proton or X-ray irradiation. Data were considered significant if * p-value < 0.05 vs. control, ** p-value < 0.05 −statistically significant differences between PdPt NPs-assisted X-ray and proton irradiation. *** p-value < 0.05 −statistically significant differences between respective cancer and normal cells, **** p-value < 0.05 −statistically significant differences between the radiosensitizing effect of PdPt (I) and PdPt (II). Adapted with permission [67]. IOPScience, Copyright 2012; Adapted with permission [70]. Elsevier, Copyright 2019; Adapted with permission [71], MDPI, Copyright 2022.

**Figure 5 nanomaterials-13-01011-f005:**
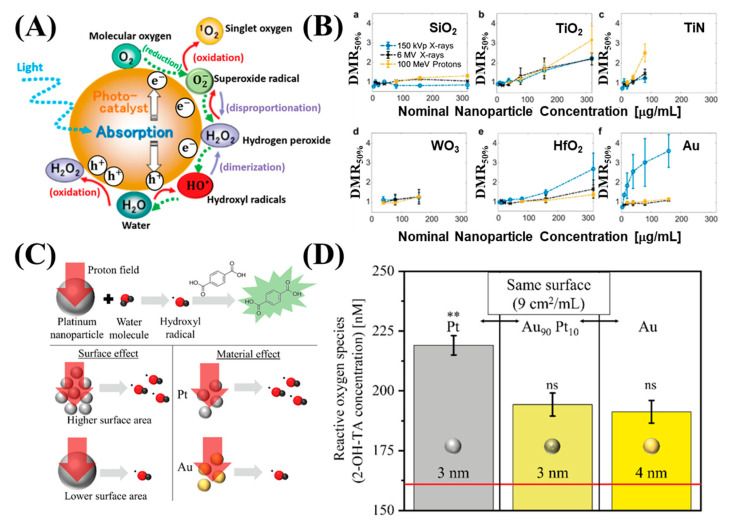
(**A**) A schematic diagram of pathways to generate reactive oxygen species in an irradiated solution containing photocatalytic NPs. (**B**) Radiation enhancement effect of (a) SiO_2_, (b) TiO_2_, (c) TiN, (d) WO_3_, (e) HfO_2_, and (f) AuNPs in HT1080 cells. (**C**) Schematic diagram of hydroxyl radical HO ^•^ generation and determination during proton irradiation of surfactant-free NPs. (**D**) Hydroxyl radical generation in Pt, Au_90_Pt_10_, and Au NPs at constant surface concentrations under proton irradiation. Asterisks show the significance of differences between 3 nm Pt NPs and Au_90_Pt_10_ and Au NPs using ANOVA analysis. Adapted with permission [77,79]. Wiley, Copyright 2020 and 2021. Adapted with permission [78]. Nature, Copyright 2022.

**Table 1 nanomaterials-13-01011-t001:** Clinical trials of the nano-radioenhancers (NREs).

Name	Particle Type/Drug	Trial ID Phase	Disease	Patients	Status	Ref.
AGuIX	Polysiloxane Gd-chelates-based NPs	NCT02820454 Phase I	Brain metastases	15	Completed	[29]
		NCT03818386 Phase II	Brain metastases	100	Recruiting	[30,31,32]
		NCT03308604 Phase I	Gynecologic Cancer	18	Recruiting	
		NCT04094077 Phase II	Brain Metastases	1	Terminated	
		NCT04789486 Phase I & II	NSCLC ^1^ Pancreatic cancer Pancreatic ductal adenocarcinoma	100	Recruiting	
		NCT04899908 Phase II	Brain cancer Brain metastases Melanoma Lung cancer Breast cancer Colorectal cancer Gastrointestinal cancer	112	Recruiting	
		NCT04881032 Phase I & II	Glioblastoma	66	Recruiting	
		NCT04784221 Phase II ^4^	Recurrent cancer previous radiation	46	Not yet recruiting	
NBTXR3	Hafnium oxide-based NPs	NCT01433068 Phase I	Adult soft tissue sarcoma	22	Completed	
		NCT02379845 Phase II & III	Adult soft tissue sarcoma	180	Completed	[33]
		NCT01946867 Phase I	HNC ^2^	63	Recruiting	[34]
		NCT02721056 Phase I & II	Liver cancer	23	Terminated	
		NCT02805894 Phase I & II	Prostate cancer	5	Terminated	
		NCT02465593 Phase I & II	Rectal cancer	32	Terminated	[35]
		NCT02901483 Phase I & II	HNSCC ^3^	12	Terminated	
		NCT03589339 Phase I	HNC Metastasis from malignant tumor of liver squamous cell cervix/skin/bladder Metastatic renal cell/triple-negative breast carcinoma	145	Recruiting	
		NCT04484909 Phase I	Pancreatic adenocarcinoma Pancreatic ductal adenocarcinoma	24	Recruiting	
		NCT04505267 Phase I	Unresectable and recurrent NSCLC Lung cancer	24	Recruiting	
		NCT04615013 Phase I	Cervical esophagus adenocarcinoma Gastroesophageal junction adenocarcinoma Thoracic esophagus adenocarcinoma	24	Recruiting	
		NCT04862455 Phase II	Metastatic and recurrent HNSCC	60	Recruiting	
		NCT04892173 Phase III	HNSCC	500	Recruiting	
		NCT05039632 Phase I & II	Advanced malignant and metastatic solid neoplasm	40	Not yet recruiting	
SPION	Iron oxide NPs/ferumoxytol	NCT04682847	Hepatic cancers	25	Recruiting	
Abraxane	albumin-bound paclitaxel	NCT00736619 Phase I	HNC	25	Completed	[36]
		NCT01921751 Phase II	Pancreatic adenocarcinoma Pancreatic cancer	20	Terminated	[37]
		NCT02394535 Phase I	Pancreatic adenocarcinoma Pancreatic cancer	25	Completed	[38]
		NCT03107182 Phase II	HPV-related squamous cell carcinoma HNSCC	76	Active, not recruiting	

^1^ NSCLC: non-small cell lung cancer; ^2^ HNC: head and neck cancer; ^3^ HNSCC: head and neck squamous cell carcinoma; ^4^ proton therapy.

## Data Availability

Not applicable.

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
