# Peer review of "Recent Advances in Metal-Based NanoEnhancers for Particle Therapy"

_nanomaterials, 2023, doi:10.3390/nano13061011_

Round 1

Reviewer 1 Report

In their article entitled " Recent advances in metal-based nanoenhancers for particle therapy", the authors offer an interesting vision of the use of metallic nanoparticles as radiosensitizing agents in the context of hadrontherapy treatments. The topic is well covered and original since most other reviews focus on the use in photon treatments which represent the large majority of what is used in the clinic. Below are my comments:

* Introduction (line 70): The authors state that proton therapy may be superior to photon treatments for a large number of clinical indications. This has long been argued in the "hadrontherapy" community. A few lines on the clinical evidence, obtained through randomized trials, should be added.

* Table 1 and main-text: Abraxane seems to me a bit out-of-scope in this review. Although it is a nano-object, I'm not sure it can be classified as a "metal-based nanoparticle". Moreover, it is used as a vehicle to increase the intracellular concentration of paclitaxel, a very different mechanism from the rest of the review. 

* In this review, the authors have highlighted the physical and chemical mechanisms of action which are well explained. However, recent studies demonstrated the importance of biological mechanisms . Thus, the authors should discuss the role played by antioxidant enzymes (such as thioredoxin reductase for example) in the radiosensitization mechanisms, which has been demonstrated for gold and iron oxide nanoparticles. 

* Section 3: transfer to the clinic. How do the authors explain that no clinical trial on the nanoparticle/proton combination are ongoing (just with photons)? 

* Section 4: Conclusion

The very interesting discussion of modified nanoparticles enabling active targeting should be taken out of the conclusion and further developed. After a quick search in the literature, I found several articles discussing these nano-objects (coupled with cetuximab in particular) and their combination with ionizing radiation. Is there a proportional relationship between the sensitizing effect and the intratumoral metal concentration? I think this should be discussed.  

Author Response

Reviewer #1

In their article entitled " Recent advances in metal-based nanoenhancers for particle therapy", the authors offer an interesting vision of the use of metallic nanoparticles as radiosensitizing agents in the context of hadrontherapy treatments. The topic is well covered and original since most other reviews focus on the use in photon treatments which represent the large majority of what is used in the clinic. Below are my comments:

Q1-1* Introduction (line 70): The authors state that proton therapy may be superior to photon treatments for a large number of clinical indications. This has long been argued in the "hadrontherapy" community. A few lines on the clinical evidence, obtained through randomized trials, should be added.

Response: As suggested by the reviewer, we have added the description of clinical evidence in the part of “Introduction”. Because benefit of protons over photons is controversial and it need more clinical evidence to proof that. In order to avoid any misunderstanding, we have modified the original manuscript. (Page 2, lines 72-79)

Included sentence

Particle therapy is thus considered, at least for a number of indications, superior to conventional RT [7-9]. Hung et al. compared clinical outcomes of proton versus photon ablative radiation therapy in patients with unresectable hepatocellular carcinoma (HCC). In their study, treatment with proton radiation therapy showed improved overall survival and decreased incidence of non-classic radiation-induced liver disease as compared with photon radiation therapy. Consistent with their results, Huang’s group compared the clinical outcomes of HCC patients, those treated with either PT or RT and found the photon group had better overall survival rate. Even though, clinical evidence demonstrating the benefit of protons over photons is still limited.

  1. Cheng, J.Y.; Liu, C.M.; Wang, Y.M.; Hsu, H.C.; Huang, E.Y.; Huang, T.T.; Lee, C.H. Hung, S.P.; Huang, B.S. Proton versus Photon Radiotherapy for Primary Hepatocellular Carcinoma: A Propensity-matched Analysis. Radiat Oncol. 2020, 15:159. doi: 10.1186/s13014-020-01605-4.

Q1-2* Table 1 and main-text: Abraxane seems to me a bit out-of-scope in this review. Although it is a nano-object, I'm not sure it can be classified as a "metal-based nanoparticle". Moreover, it is used as a vehicle to increase the intracellular concentration of paclitaxel, a very different mechanism from the rest of the review.

Response: Thank you for bringing this to our attention. We apologize for the mistake made by negligence in the original version. Not all of NPs in this manuscript are metal-based NPs. As we maintained in section 3, Abraxane is an albumin-bound form of paclitaxel, it cannot be classified as a "metal-based nanoparticle. In Table 1, we list Clinical trials of the nano-radioenhancers (NREs).The radiosensitizing effect of paclitaxel on cancer cells is due to the down-regulated expression of PRC1 and cyclin B2, resulting in inhibition of mitotic spindle formation and cell necrosis. It belongs to biological effect of radiosensitization. In addition, some research groups called Abraxane as a radiosensitizer in Abraxane related publications. In view of this, we classified it to NREs.   

Q1-3* In this review, the authors have highlighted the physical and chemical mechanisms of action which are well explained. However, recent studies demonstrated the importance of biological mechanisms . Thus, the authors should discuss the role played by antioxidant enzymes (such as thioredoxin reductase for example) in the radiosensitization mechanisms, which has been demonstrated for gold and iron oxide nanoparticles.

Response: Thanks for the reviewer’s comment. Biological effect as well as physical and chemical mechanisms plays a critical role in radiosensitizing effect. A number of studies have shown that the intracellular NPs contribute significantly to dose enhancement, whereas the extracellular or membrane-bound NPs do not cause significant radiosensitization. We have list three key biological pathways (1) oxidative stress, (2) cell cycle disruption, and (3) DNA repair inhibition in our revised manuscript to support the biological contribution to radiosensitization.

Included sentence (Page 12, lines 375-380)

In addition to physical and chemical mechanisms, biological effect contributes signifi-cantly to radiosensitization. Intracellular accumulation of NPs may affect antioxidant enzymes (such as thioredoxin reductase and glutathione peroxidase) expression and change the balance of oxidative stress. Besides, NPs sensitize cancer cells to radiation by causing cell cycle disruption and DNA repair inhibition should not be ignored.

Q1-4* Section 3: transfer to the clinic. How do the authors explain that no clinical trial on the nanoparticle/proton combination are ongoing (just with photons)?

Response: NERs in proton therapy is an important and emerging area. Based on “ClinicalTrials.gov” and “EU Clinical Trials Register” websites to date, three clinical trials on the nanoparticle/proton combination have been identified: two for AGuIX (2020-003671-17/EMA and NCT04784221/FDA) and one for NBTXR3 (NCT04834349/FDA). The status of NCT04834349/FDA is “Withdrawn” and the status of         NCT04784221/FDA is “Active, not recruiting”. We have brief descripted their status in section 3.

Q1-5* Section 4: Conclusion

The very interesting discussion of modified nanoparticles enabling active targeting should be taken out of the conclusion and further developed. After a quick search in the literature, I found several articles discussing these nano-objects (coupled with cetuximab in particular) and their combination with ionizing radiation. Is there a proportional relationship between the sensitizing effect and the intratumoral metal concentration? I think this should be discussed. 

Response: Thanks for the reviewer’s comment. Indeed, engineered MNPs enable active targeting not only improves the intratumoral NPs concentration, the specific targeting (such as mitochondria and nucleus) also strengthens damage in DNA. The description of how engineered MNPs affect the intratumoral metal concentration and enhances radiosensitizing effect have been added in our revised manuscript. (Page 12, lines 369-375)

Before modification:

Engineered MNPs that are functionalized with moieties able to actively target a tumor or its microenvironment have the potential to improve upon the efficacy and accuracy.

After modification:

Engineered MNPs that are functionalized with moieties able to actively target a tumor or its microenvironment have the potential to improve upon the efficacy and accuracy. Furthermore, ROS generated from the radiolysis of water deposits in the immediate vicinity of the NERs and considers with their short life time in cells, the travelling dis-tance of ROS is not more than 6 nm on average. Therefore ROS formation closes to cell nuclei or mitochondria may play a significant role in the radiosensitisation capacity.

Reviewer 2 Report

The review from Chuang et al. details the recent advances in radiotherapy regimens for cancer treatment, with a focus on the use of particle therapy as an advanced type of radiotherapy. The review presents the efforts that have been made to improve tumor targeting by concentrating the radiation effects via a combination of nanomaterials and particle therapies. The authors summarize the unique properties of metallic nanoparticles (MNPs) such as strong X-ray absorption cross-sections and catalytic activity, and their use as nano-radioenhancers (NREs) for radiotherapy. Putative mechanisms involved in NRE therapy and the experimental results in in vitro and in vivo models are presented, together with their potential translation into clinical practice, with pertinent examples of several metal-based NREs and undergoing/completed clinical studies and their findings. The authors also discussed challenges for the development of NREs for PT and proposed a roadmap to pursue future studies to strengthen the interplay of particle therapy and nanomedicine. The review is valuable and well written. I recommend publication with the minor corrections indicated below:

Line 2: “Over the past century, radiotherapy (RT) has 52 emerged as the main treatment modality for cancer that more than two-thirds receive RT 53 alone or in combination with other treatments in their treatment history” please rephrase

Line 54: “In conventional 54 RT, photon beam is mainly used that can induce cancer cell damage without surgical incisions.” Please rephrase

Line 57: please replace “unstoppable” with “irreversible”, or equivalent

Line 85; “correct “locat”

Line 165: should read “CT-26 tumor-bearing mice were treated”

Line 230: Figure 3B is impossible to read and its quality should be improved; Figure 3 resolution should be improved.

Reviewer 3 Report

The authors have provided a history of radiotherapy and its link to the new methods involving nanoparticles (mainly magnetic) and the benefits that they bring. Overall the paper is intersting and the authors did a good job at explaining why certain studies were important and what advantages the new devolopment brings. An overall correction for the english is required along with several minor to moderate issues regarding the readability of the paper as follows:

-please indicate figures more precisely in text.... example (Figure 1a   or figure 1b)

-once an abbreviation is given at the first use please use it throughout. Many times abbreviations are used only sometimes (ex. Photon therapy)

-Lines 94-95 regarding Nanoparticle circulation this is a huge field and these 2 citations seem lacking in their background of such a vast amount of info. Please increase the citations to better cover the topic

-similarly as the previous comment, please be careful/specific when making blanket statements i.e"nanoparticles have long circulation" which nanoparticles? metal? others? because each is very different depending on the composition.

-in table 1 please separate the citations into a separate column. also watch out for "/" leading to words being cut in half between lines

-in vivo / in vitro should be italicized

-Figure 2 the difference between e+ and e-  as well as large and small impact parameters is indestinguishable. please change color or make them more identifiable.

-Please define what is "Z" for those not in the field

-Figure 3 pro-ton is hyphenated is this correct?

-Figure 3 and Figure 4 have some parts that would be very hard to read without zooming in (digital) please ensure that they are legible in the final draft

Reviewer 4 Report

Engñish must be revised in deep.

The sentence "Over the past century, radiotherapy (RT) has 52 emerged as the main treatment modality for cancer that more than two-thirds receive RT 53 alone or in combination with other treatments in their treatment history." needs references

The sentence "SOBP imparts a good dosimetric distribution to particle therapy, which maximizes the 69 efficiency of killing of tumor tissues and protects surrounding normal tissues around to a 70 great extent." needs references

The paragraph "Although particle therapy has significant advantages over conventional RT for treat-83 ing cancer by reducing the radiation damage to surrounding normal tissues, the radiation 84 dose deposited by the particle beam in normal tissues which locat in the path before the 85 tumor is unavoidable, which may cause normal tissue damage." presents repeated informatuion as well as needs references. Please correct it. 

The sentence "Of these, metal-91 based nanomaterials or “nano-radioenhancers” (NREs) have attracted significant atten-92 tion in radiation oncology because of their strong photoelectric absorption coefficients for 93 high atomic number metallic elements." needs references.

The sentence "In addition, nanomaterials circulate in the body 94 for a long time and achieve tumor-specific accumulation through targeted biomolecules 95 or the increased permeability and retention effects" needs correction (Enhanced Permeability and Retention (EPR) effect). The EPR must be described in deep (also the controversy surrounding it). DOI:10.2174/1381612821666150820100812  DOI: 10.1039/d1bm01398j

The paragraph "Actually, since the phys-102 ical mechanism is different from photon RT and particle therapy, there has been an in-103 creasing number of studies focused on the potential of particle therapy combined with 104 specific NREs, particularly involving gadolinium, and hafnium." needn references

The sentence "The basic rationale for using nanomaterials as NREs results from the physical dose 118 enhancement that occurs following radiochemical and biological reactions in the targeted 119 tissue." needs references

The sentence "The physical dose enhancement is caused by the generation of secondary X-rays, 120 photoelectrons, and Auger electrons." needs references. The reviewer recommends a Figure explaining this physical process. 

Some of these elements "In the coming years, 130 other sophisticated NPs composed of heavy elements, such as titanium (Ti, Z = 22) [33,34], 131 iodine (I, Z = 53 )[35,36], gadolinium (Gd, Z = 64) [37-39], hafnium (Hf, Z = 72) [40-42], and 132 bismuth (Bi, Z = 83) [43,44]" have been previouly mentioned. Also, the paragraph needs references.

The sentence "Based on these studies, kV X-133 rays as a radiation modality can use the high photoelectric absorption cross-section and 134 the consecutive release of secondary electrons (including Auger electrons and photoelec-135 trons) from NREs as an advantage to further enhance the radiation effects." needs references.

"measured the first radiosensitization effect result-154 ing from MNPs for cells irradiated with a proton beam." MNPs have not been previously described

Could authors explain why in the previous sentence they describe AuNPS and later they write again gold nanoparticles? "AuNPs exhibited a significant increase in radiosensitization with CIBT. 178 Kaur et al. prepared gold nanoparticles coated with glucose."

It would be really helpfull if authors describe all SER in a table, in order to properly compared them. "The sensitization enhance-185 ment ratio (SER) was 1.35-1.80 for hepatocellular carcinoma cells."

Please be consistent nanoparticles or NPs "In a recent study, Abdul Rashid et al. com-190 pared the molecular effects induced by Fe, Au, Pt, and Bi nanoparticles under the expo-191 sure of 150 MeV protons."

The reviewer believe that one reference is missing "Recently, multicomponent metallic NPs were used to improve 198 radiotherapy for simulated proton irradiation against cancer cells. Klebowski et al. syn-199 thesized AuNPs decorated with PtNPs and PdNPs by a green chemistry method using 200 gallic acid. The large surface of fancy shaped bimetallic NPs ensures a large contact area 201 with the cells and results in an increased amount of ROS destruction in cancer cells. The 202 administration of 30 nm PtAuNPs and PdAuNPs as potential radiosensitizers in PT of 203 colorectal cancer showed that this combined approach resulted in a significant inhibition 204 of cancer cell proliferation and viability, whereas normal cells were less affected during 205 treatment. Subsequently, the same group designed two bimetallic PtPdNP structures, 206 nano-alloy and core-shell, and checked the radiosensitizing properties. In this experiment, 207 nano-alloy PtPdNPs exhibited superior radiosensitization in simulated proton irradiation 208 compared with their PdPt core-shell counterparts, suggesting that the presence of Pd at-209 oms on the surface of these NPs imparts better radiosensitizing effects than Pt atoms (Fig-210 ure 3) [63]."

The sentence "In recent years, the improvement and development of nanoparticles has rapidly 251 evolved, driven by overcoming the limitations of free therapeutics, navigating biological 252 barriers, and even the clinical failures of current drugs." needs references

Section 3 lacks of references. (only 3 references)

Could authors hypothesize some of the key parameters? "Understanding key parameters and underlying mechanisms of action involved in 333 the radiosensitization effects of NPs remains an issue for developing new NREs."

Authors would like to describe larger  surface area-to-volume ratio? "Higher 334 NP surface area" 

Please check English, as well as, EPR effect. Would only NPs  accumulation cure cancer? "The use of untargeted MNPs results in inefficient accumulation in the tumor through en-338 hanced permeability and the retention effect is insufficient to cure cancer." 

"Other routes, 339 such as intra-tumoral, and intra-operative delivery, should be considered. However, these 340 routes can only be applied to a limited number of eligible cancers." Hence, authors must re-write this paragraph, and describe the smart design of NPs to increase intratumoral concentration after intravenous administration

Reviewer recommends to re-write the conclusion. 
